# Photon- and Singlet-Oxygen-Induced Cis–Trans Isomerization of the Water-Soluble Carotenoid Crocin

**DOI:** 10.3390/ijms241310783

**Published:** 2023-06-28

**Authors:** Franco Fusi, Giovanni Romano, Giovanna Speranza, Giovanni Agati

**Affiliations:** 1Department of Experimental and Clinical Biomedical Sciences “Mario Serio”, University of Florence, Viale G. Pieraccini, 6, 50139 Florence, Italy; franco.fusi@unifi.it; 2Department of Chemistry, University of Milan, Via Golgi 19, 20133 Milan, Italy; giovanna.speranza@unimi.it; 3“Nello Carrara” Institute of Applied Physics (IFAC), National Research Council (CNR), Via Madonna del Piano 10, 50019 Sesto Fiorentino, Italy; g.agati@ifac.cnr.it

**Keywords:** crocin, differential absorbance, photoisomerization, quantum yields, quenching, singlet oxygen, Z-E isomers, antioxidant, crocin based assay CBA

## Abstract

Studying the cis–trans isomerization process in crocin (CR), one of the few water-soluble carotenoids extracted from saffron, is important to better understand the physiological role of cis-carotenoids in vivo and their potential as antioxidants in therapeutic applications. For that, cis–trans isomerization of both methanol- and water-dissolved CR was induced by light or thermally generated singlet oxygen (^1^O_2_). The kinetics of molecular concentrations were monitored by both high-performance liquid chromatography (HPLC) and non-destructive spectrophotometric methods. These last made it possible to simultaneously follow the cis–trans isomerization, the possible bleaching of compounds and the amount of thermally generated ^1^O_2_. Our results were in accordance with a comprehensive model where the cis–trans isomerization occurs as relaxation from the triplet state of all-trans- or 13-cis-CR, whatever is the way to populate the CR triplet state, either by photon or ^1^O_2_ energy transfer. The process is much more (1.9 to 10-fold) efficient from cis to trans than vice versa. In H_2_O, a ^1^O_2_-induced bleaching effect on the starting CR was not negligible. However, the CR “flip-flop” isomerization reaction could still occur, suggesting that this process can represent an efficient mechanism for quenching of reactive oxygen species (ROS) in vivo, with a limited need of carotenoid regeneration.

## 1. Introduction

Carotenoids are well known natural pigments utilized by photosynthetic systems in the light-harvesting as well as in the photoprotection mechanisms [1]. They easily undergo cis–trans isomerization; however, the biological functional role of the resulting different configurations has not been completely elucidated yet.

Normally carotenoids occur in nature as the all-trans isomer. However, cis configurations have also been isolated from several vegetables [2,3,4] and the *Dunaliella bardawil* green alga [2]. The differential localization of all-trans and 15-cis isomers of β-carotene in the photosynthetic apparatus of the *Rhodospirillum rubrum* S1 bacterium indicates a specific biological function of these compounds linked to their geometrical structure. In the reaction center, the presence of the 15-cis configuration, with a longer conjugated chain, is suggested to have a more efficient photoprotection role with respect to the all-trans isomer in dissipating the excess of energy. While the all-trans configuration, with a shorter conjugated chain, in the light-harvest complex provides an optimized excitation energy transfer to chlorophyll [5]. Therefore, cis–trans carotenoids are important actors in light absorption and membrane protection against ROS (singlet oxygen and free radicals) [1].

Cis-carotenes are also involved as molecular signals to control plastid activities and plant development [6], while the cis/trans isomer ratio in food has been seen to increase during processing [7]. On the other hand, the bioavailability of carotenoid cis-isomers and their accumulation in human serum and tissues was found to be higher than the all-trans-isomers [8,9]. This is likely due to the decreased tendency of cis-isomers to form aggregates and crystals [10].

The high efficiency of carotenoids in quenching singlet oxygen focused large interest on these molecules as antioxidant agents [11]. This property can also explain the in vivo observation of a positive role of carotenoids in the prevention of cancer [12], in delaying the progress of neurodegenerative diseases [13] and in providing other numerous health benefits [14].

Since the ^3^O_2_ excitation energy to produce ^1^O_2_ is 94.5 kJ⋅mol^−1^, ^1^O_2_ can be easily produced in vivo by natural sensitizers (e.g., chlorophyll) and its production is followed by oxidation of several cellular components. This process can be significantly reduced by carotenoids able to physically quench ^1^O_2_ by energy transfer [15] and also inducing the cis–trans isomerization [3].

The cis–trans interconversion is a well-known process that can be induced by light, directly or through a photosensitizer [16], as well as by high temperatures (>70 °C) [16,17].

High-performance liquid chromatography (HPLC) along with time-resolved Raman and absorption spectroscopy provided an accurate quantitative evaluation of β-carotene photoisomerization. The results showed that: (1) the cis–trans isomerization occurs as relaxation of the triplet excited state even in the case of direct photoisomerization; (2) the all-trans and 15-cis isomers have a common intermediate triplet state; (3) the isomerization of 15-cis is an order of magnitude more efficient than isomerization of all-trans [16,18,19].

Further evidence about the triplet-state-mediated photoisomerization route was also reported for zeaxanthin [20].

The isomerization reaction following ^1^O_2_ quenching by carotenoids is much less studied.

More than fifty years ago, it was first observed that ^1^O_2_ can sensitize the 15-cis to all-trans-β-carotene isomerization [21]. In that study, based only on absorption spectroscopy, no appreciable inverse trans-to-cis process was observed. Much later, Heymann et al. [3] clearly showed that ^1^O_2_ can produce different cis isomers, starting the reaction from the all-trans configuration in both lycopene and β-carotene.

Several authors reported data on the physical and chemical ^1^O_2_-quenching rate constants of several trans and cis carotenoids [15,22,23,24,25]. However, the reaction models used did not consider the potential contribution to the quenching rates of the cis–trans isomerization.

Most of the carotenoid isomerization processes have been extensively studied in non-polar organic solvents, in which they possess a high solubility. In this environment, carotenoids are able to quench singlet oxygen physically rather than chemically [22].

In some studies, conducted in aqueous media by using reverse micelle systems, a large difference of four orders of magnitude between the physical and chemical ^1^O_2_ quenching rate constants of carotenoids was still found [26,27].

Information on the photophysical and chemical properties of water-soluble carotenoids is relevant to the understanding of the protective role they play in vivo in the photodynamic oxidation of biological systems.

In the presence of high concentrations of ROS, the organism can use exogenous antioxidants supplied with the diet or through pharmaceutical products. To characterize an antioxidant compound, it is necessary to consider the mechanisms involved and therefore both the activity and the antioxidant capacity. Antioxidant activity refers to the rate constant of a reaction between an antioxidant and an oxidant. Antioxidant capacity is a measure of the amount of a certain free radical captured by an antioxidant sample.

In the literature [28], an in vitro method (CBA- Crocin Based Assay) is presented for measuring the antioxidant and prooxidant capacity of nutritional supplements, pharmaceuticals and biological samples, based on a hydro-soluble carotenoid. 

Crocin (CR) [29] is the major pigment of saffron, the spice obtained from the dried stigmata of *Crocus sativus* L., to whom it confers the characteristic yellow color. Being a water-soluble carotenoid, it is a unique antioxidant. It is used in the treatment of many diseases [30] and for cytoprotective properties, including systemic [31] or local [32] anti-inflammatory effect.

Only few studies on photochemical and ^1^O_2_-induced reactions in water-soluble carotenoids, such as 8,8′-diapocarotene-8,8′-dioic acid (crocetin) [23,33], its digentiobiosyl ester (CR) [23,34,35] and a series of 6,6′-diapocarotenoids [25,34], have been reported.

In the present paper, we report a complete investigation on the cis–trans isomerization of all-trans- and 13-cis-CR (Figure 1) induced by photon or by energy transfer from singlet oxygen in methanol as well as aqueous solutions. A spectroscopic method for the contemporary detection of ^1^O_2_ production, carotenoid bleaching and isomerization is proposed. Additional quantitative information on the trans/cis isomer concentrations during the reaction processes were obtained by HPLC analysis.

## 2. Results

Based on the actual knowledge about the way to induce the cis–trans isomerization in carotenoid molecules [16,20], we schematized the possible processes occurring in CR, as reported in Figure 2.

### 2.1. Kinetics of the all-trans↔13-cis Photoreaction

According to the scheme reported in Figure 2, the kinetics of the all-trans↔13-cis photoreaction during monochromatic irradiation at the λ_i_ wavelength, neglecting any photobleaching, is described by the following equation:(1)Ct˙=−AtCt+KtICC1t+Kt’C3t+KtC3cC1t˙=AtCt−KtIC+KtISCC1tC3t˙=KtISCC1t− Kc+Kt’C3tCc˙=−AcCc+KcICC1c+Kc’C3c+KcC3tC1c˙=AcCc−(KcIC+KcISC)C1cC3c˙=KcISCC1c− Kt+Kc’C3cCt+CcC0=1
where $C_{t}$ and $C_{c}$ denote the concentrations of the all*-trans* and 13*-cis* isomers in the ground state, respectively, and $C_{0}$ is the initial concentration of the solution. The 1, 3 superscript numbers of concentrations mean singlet- and triplet-excited states; the dot apices indicate the time derivatives of populations; and the light excitation of ground-state molecules to the singlet excited states is driven by the rate constants.
(2)$$A_{t}=\sigma_{t}\left(\lambda_{i}\right)\cdot F;\;A_{c}=\sigma_{c}\left(\lambda_{i}\right)\cdot F$$
where $\sigma_{t}\left(\lambda_{i}\right)$ and $\sigma_{c}\left(\lambda_{i}\right)$ represent the absorption cross-sections of all*-trans* and 13*-cis*, respectively, and *F* denotes the incident photon fluence rate. 

Equations (1) and (2) are valid in the limit of an optically thin solution, αl≪1, with α and *l* denoting the absorption coefficient and the thickness of the solution, respectively.

Over the thickness *l* of the solution with the absorption coefficient α, the spatial average of *F* is
(3)$$\langle F\rangle =\frac{F}{l}\int_{0}^{l}e^{-\alpha x}dx=F\cdot \frac{\left(1-e^{-\alpha l}\right)}{\alpha l}$$

The averaged *F* value was considered in the following equations to give a more accurate determination of the photoisomeriation quantum yields (Equation (13)), even if in our case *F* variations along the light path are of the order of 5–15% and could be neglected. Within the steady-state approximation, C1t˙= C3t˙ =C1c˙ = C3c˙ = 0, we have that
(4)C1t=AtCtKtIC+KtISC ; C1c=AcCcKcIC+KcISCC3t=AtCt·KtISC(KtIC+KtISC)·(Kc+Kt’)  ; C3c=AcCc·KcISC(KcIC+KcISC)·(Kt+Kc’)
that combined with Equation (1) lead to
(5)Ct˙=−YcΦtISCAtCt+YtΦcISCAcCc=−ϕcAtCt+ϕtAcCcCc˙=−YtΦcISCAcCc+YcΦtISCAtCt=−ϕtAcCc+ϕcAtCt
with
(6)$$Y_{c}=\frac{K_{c}}{K_{c}+K_{t}^{’}}\;;\;Y_{t}=\frac{K_{t}}{K_{t}+K_{c}^{’}}$$
defined as the yields of isomerization from trans to cis ($Y_{c}$) and vice versa ($Y_{t}$), respectively, and
(7)$$\Phi_{t}^{ISC}=\frac{K_{t}^{ISC}}{\left(K_{t}^{IC}+K_{t}^{ISC}\right)}\;;\;\Phi_{c}^{ISC}=\frac{K_{c}^{ISC}}{\left(K_{c}^{IC}+K_{c}^{ISC}\right)}$$
are the triplet-state quantum yields through intersystem crossing for all-trans- and 13-cis-CR, respectively.

We can then define the quantum yields of all-trans to 13-cis photoisomerization, and vice versa from 13-cis to all-trans, as the product of the triplet-state quantum yields by the yield of isomerization from the triplet state:(8)$$\phi_{c}=Y_{c}\Phi_{t}^{ISC}\;;\;\phi_{t}=Y_{t}\Phi_{c}^{ISC}$$

Starting the irradiation, at $\lambda_{i}$, of a CR solution, the time evolution of the relative concentration of CR isomers, defined as $c_{t,c}=C_{t,c}/C_{0}$, which satisfy the initial conditions, at t = 0, $c_{t}$(0) = 1 and $c_{c}$(0) = 0 are given as solutions of the system (5):(9)$$c_{t}\left(t\right)=c_{t}\left(\infty \right)+c_{c}\left(\infty \right)\cdot e^{-\frac{t}{\tau }}\;c_{c}\left(t\right)=c_{c}\left(\infty \right)(1-e^{-\frac{t}{\tau }})$$
where
(10)$$c_{t}\left(\infty \right)={\left(1+\frac{Y_{c}\Phi_{t}^{ISC}\sigma_{t}}{Y_{t}\Phi_{c}^{ISC}\sigma_{c}}\right)}^{-1}={\left(1+\frac{\phi_{c}\sigma_{t}}{\phi_{t}\sigma_{c}}\right)}^{-1}\;c_{c}\left(\infty \right)={\left(1+\frac{Y_{t}\Phi_{c}^{ISC}\sigma_{c}}{Y_{c}\Phi_{t}^{ISC}\sigma_{t}}\right)}^{-1}={\left(1+\frac{\phi_{t}\sigma_{c}}{\phi_{c}\sigma_{t}}\right)}^{-1}$$
are the relative concentrations of the two isomers at the photoequilibrium, with
(11)$$\tau^{-1}=\langle F\rangle \left(\phi_{t}\sigma_{c}+\phi_{c}\sigma_{t\;}\right)$$
and combining the two expressions in Equation (10), we obtain
(12)$$\frac{\phi_{t}}{\phi_{c}}=\frac{c_{t}\left(\infty \right)}{c_{c}\left(\infty \right)}\;\cdot \;\frac{\sigma_{t}}{\sigma_{c}}=\frac{c_{t}\left(\infty \right)}{c_{c}\left(\infty \right)}\;\cdot \;\frac{\varepsilon_{t}}{\varepsilon_{c}}\left(\mathrm{being}\;\varepsilon =\frac{Na}{ln10}\sigma \right)$$

At the initial stage of the all-trans to 13-cis photoisomerization and vice versa, around t = 0, from Equation (5) we can calculate the quantum yields
(13)ϕc=−Ct˙σtFCt0=−ct˙σtF=τ−1cc∞σtFϕt=−Cc˙σcFCc0=−cc˙σcF=τ−1ct∞σtF
where the right-end expressions were obtained by calculating the first derivative in the origin of Equation (9).

In the trans–cis photoisomerization process, and in the absence of any photobleaching, the absorption coefficient of the CR solution as a function of time and wavelength is given by
(14)$$\alpha \left(t,\lambda \right)=C_{0}\left(\sigma_{t}\left(\lambda \right)\cdot c_{t}\left(t\right)+\sigma_{c}\left(\lambda \right)\cdot c_{c}\left(t\right)\right)$$
where $C_{0}$ represents the initial concentration of CR molecules.

As a result, during the irradiation of a solution of all*-trans*-CR (or 13*-cis*-CR), the difference absorption coefficient Δ*α*(*t*, *λ*) evolves concurrently with $c_{c}$ (or$\cdot c_{t}$):(15)$$\Delta \alpha \left(t,\lambda \right)=\alpha \left(t,\lambda \right)-\alpha \left(0,\lambda \right)=\alpha \left(0,\lambda \right)\cdot \left(\frac{\sigma_{c}\left(\lambda \right)}{\sigma_{t}\left(\lambda \right)}-1\right)\cdot c_{c}\left(t\right)=\alpha \left(0,\lambda \right)\cdot \left(\frac{\sigma_{t}\left(\lambda \right)}{\sigma_{c}\left(\lambda \right)}-1\right)\cdot c_{t}\left(t\right)\;\;$$

Then, considering Equation (9), by fitting the time evolution to the photoequilibrium of Δ*α* at specific wavelengths makes it possible to determine the parameters, τ and $c_{c}\left(\infty \right)$ (or$\cdot c_{t}\left(\infty \right)$), needed to calculate the photoisomerization quantum yields, according to Equation (13).

From Equation (15), it is worth noting that isosbestic points (Δα=0) occur at wavelengths for which $\sigma_{t}=\sigma_{c}$.

### 2.2. Absorption Spectral Properties

The extinction coefficient spectra of all-trans- and 13-cis-CR in methanol (MeOH) and aqueous solutions are reported in Figure 3A. In MeOH, the all-trans isomer was characterized by the main peaks at 432 and 458 nm and a shoulder at around 409 nm. The 13-cis isomer presented visible bands slightly shifted at lower wavelengths, 426 and 452 nm, and a marked band at 320 nm. The extinction coefficient (ε) of 13-cis-CR at the main peak was about 30% lower than the all-trans-CR, at 64.7 kM^−1^ cm^−1^ versus 89.7 kM^−1^ cm^−1^.

In H_2_O, the main peaks occurred at 442 (ε = 85.7 kM^−1^ cm^−1^) and 432 nm (ε = 51.3 kM^−1^ cm^−1^) for all-trans-CR and 13-cis-CR, respectively, slightly shifted towards longer wavelengths with respect to the methanolic solution. The “cis” peak was present at 326 nm.

### 2.3. Direct Photoisomerization

Because of the above absorption properties of isomers, the photon-induced conversion of all-trans-CR to 13-cis-CR determined the typical difference absorbance (DA) spectrum (irradiated minus unirradiated) depicted in Figure 3B,C, with negative bands in the visible and a positive band in the UV spectral region. Photoisomerization starting from 13-cis-CR produced difference absorbance spectra specular in shape to those starting from all-trans-CR (Figure 3B,C). The reported spectra were obtained after the photoequilibrium was reached. 

Both spectra showed isosbestic points at 266 and 380 nm. Variation of absorbance at these wavelengths could be used to check if some other products or bleaching of the molecules occurred. The wavelengths of maximal molar extinction difference at 321 and 463 nm could instead be used to follow the change of 13-cis-CR and all-trans-CR concentrations starting from pure compound solutions.

Figure 4 shows the time evolution of the DA spectra peak values at 321 and 463 nm of methanolic solutions, both aerated (saturated with molecular oxygen) or deoxygenated (by argon bubbling), exposed to a constant photon fluence rate at 488 nm, starting from all-trans-CR. The two peaks changed in opposite direction, indicating the formation of 13-cis-CR and the disappearance of the all-trans isomer.

The experimental data of Figure 4 were accurately fitted by an exponential curve. Hence, the formation of the 13-cis isomer followed a first-order kinetics up to a photoequilibrium. Analogously, the time evolution of difference absorbances during photoisomerization starting from the 13-cis-CR are reported in Appendix A.

The presence of oxygen did not appreciably affect the spectral values at the isosbestic points; consequently, no reaction of photooxidation was observed. Furthermore, the photoisomerization reaction was faster in the presence of oxygen than without oxygen.

According to Equation (15), the *13-cis*-CR concentration at any time t was proportional to the DA and could be evaluated once the σcis/σtrans ratio was known. 

The quantum yields of photoisomerization evaluated by using Equation (13) for irradiated solutions of all-trans- ($\phi_{c,m}$) or 13-cis-CR ($\phi_{t,m}$), defined as directly measured, are reported in Table 1. These values were then used in Equation (12) to predict the quantum yields of the back reactions, cis to trans ($\phi_{t,p}$) and trans to cis ($\phi_{c,p}$), respectively, by the model used to describe the kinetics of CR photoisomers. The comparison between directly measured (m) and predicted (p) quantum yields is useful to check the accuracy of the photoisomerization model.

These calculations were performed by using the DA values at both the 463 and 321 nm. Although there was a tendency to obtain higher values of quantum yields by following the cis peak at 321 nm, no significant difference between the average values calculated at the two spectral bands (*p* > 0.5, according to Welch’s *t*-test) was observed. Consequently, we grouped the two data sets, and report the resulting average values in Table 1. In the same Table, the photoequilibrium concentration of 13-cis-CR determined by the spectrophotometric analysis is reported.

In Ar-saturated MeOH solutions, the $\phi_{c,m}$ in the trans → cis photoisomerization was 4.4-times lower than $\phi_{t,m}$ in the cis → trans photoreaction. In addition, $\phi_{c,m}$ was more than one order of magnitude (13.3 times) lower than $\phi_{t,p}$, as predicted by the photoisomerization model. Starting from the 13-cis-CR (cis → trans) still produced higher values for the measured $\phi_{t,m}$ than the predicted $\phi_{c,p}$ ($\frac{\phi_{t,m}}{\phi_{c,p}}$ ratio = 6.7).

On the other hand, the $\phi_{c,m}$ directly measured was significantly (*p* = 0.038) higher than that predicted by Equation (12) starting from the 13-cis-CR ($\phi_{c,p}$) and the $\phi_{t,m}$ measured was three times lower than that predicted ($\phi_{t,p}$) (*p* < 0.001).

In oxygen-saturated MeOH solutions, both cis and trans quantum yields wer higher than those in Ar-saturated solutions, 2.3 and 5.5 for the measured $\phi_{c,m}$ and $\phi_{t,m}$, respectively. The presence of oxygen, therefore, favored the photoisomerization of crocins.

For both solutions, the 13-cis-CR concentration of photoequilibrium starting from 13-cis-CR was higher (52–74%) than that obtained by starting from all-trans-CR.

In Ar-saturated water solutions, the measured $\phi_{t,m}$ starting from 13-cis-crocin was one of order of magnitude higher than the measured $\phi_{c,m}$ starting from all-trans. The predicted $\phi_{t,p}$ was even much higher (24 times) than the measured $\phi_{c,m}$.

On the other hand, the directly measured $\phi_{c,m}$ was not significantly (*p* = 0.187) different than that predicted by Equation (12) starting from the 13-cis-CR, while the measured $\phi_{t,m}$ was significantly (*p* = 0.036) lower than that predicted (*p* < 0.001).

The 13-cis-CR concentration of photoequilibrium starting from 13-cis-CR was 1.5-times higher than that obtained by starting from all-trans-CR.

### 2.4. Kinetics of the CR–^1^O_2_ Reactions

All of the processes occurring during the reactions between CR and ^1^O_2_ (Figure 2) can be described by the following Equations.
(16)EP→KD Naph+O12O12→KSO32Ct+O12→KRt productsCt+O12→KQt C3t+O32C3t→KcCcC3t→Kt’CtCc+O12→KRc productsCc+O12→KQc C3c+O32C3c→KtCtC3c→Kc’Cc
where EP is the endoperoxide that produces the naphthalenic acid (Naph) and ^1^O_2_ with the thermal dissociation rate $K_{D}$ (see Materials and Methods section); $C_{t}$ and $C_{c}$ represent the concentration of all-trans- and 13-cis-CR in the stationary state (or triplet state: C3t and C3c). $K_{Q_{t,c}}$ and $K_{R_{t,c}}$ are the bimolecular rate constants for physical and chemical quenching of ^1^O_2_, respectively. The deexcitation rates of the CR triplet states are indicated by $K_{t}$ and $K_{c}$ for the trans↔cis isomerization processes and by $K_{t}^{’}$ and $K_{c}^{’}$ for the internal relaxation processes. The rate constants for the natural decay of ^1^O_2_, $K_{S}$, was determined to be 1.43 × 10^5^ s^−1^ in MeOH [36] and 2.4 × 10^5^ s^−1^ [36] in H_2_O [37].

The time evolution of concentrations is given by
(17)Ct˙=Kt’C3t+KtC3c−(KQt+KRt)CtO12Cc˙=Kc’C3c+KcC3t−(KQc+KRc)CcO12C3t˙=KQtCtO12− Kc+Kt’C3tC3t˙=KQtCtO12− Kc+Kt’C3tO12˙=KDEPYSO−KS+KQt+KRt Ct+(KQc+KRc) CcO12

Within the steady-state approximation, C3t˙=C3c˙=0, we have that
(18)C3t=KQtCtO12Kc+Kt’ ; C3c=KQcCcO12Kt+Kc’
that combined with Equation (17) lead to
(19)Ct˙=−YcKQt+KRtCt+YtKQcCcO12Cc˙=−YtKQc+KRcCc+YcKQtCtO12
with
(20)$$Y_{c}=\frac{K_{c}}{K_{c}+K_{t}^{’}}\;;\;Y_{t}=\frac{K_{t}}{K_{t}+K_{c}^{’}}$$
as defined above in the photoisomerization section (Equation (6)).

At the initial stage of the C_t_–^1^O_2_ reaction, around t = 0, from the last of Equation (17) we have that
(21)O12=KDEP0YSOKS+KQt+KRtCt0
that used in Equation (19) allows to calculate
(22)$$Y_{c}=\dot{-C_{t0}}\cdot \frac{K_{s}+\left(K_{Q_{t}}+K_{R_{t}}\right)C_{t0}}{K_{Q_{t}}C_{t0}\cdot K_{D}EP_{0}Y_{SO}}-\frac{K_{R_{t}}}{K_{Q_{t}}}$$

Analogously, for the C_c_–^1^O_2_ reaction we obtain
(23)$$Y_{t}=\dot{-C_{c0}}\cdot \frac{K_{s}+\left(K_{Q_{c}}+K_{R_{c}}\right)C_{c0}}{K_{Q_{c}}C_{c0}\cdot K_{D}EP_{0}Y_{SO}}-\frac{K_{R_{c}}}{K_{Q_{c}}}$$

At the steady state, being $\dot{C_{t}}=\dot{C_{c}}=0$, Equation (19) gives
(24)$$\left(Y_{c}K_{Q_{t}}+K_{R_{t}}\right){\left[C_{t}\right]}_{\infty }=Y_{t}K_{Q_{c}}{\left[C_{c}\right]}_{\infty }\;\left(Y_{t}K_{Q_{c}}+K_{R_{c}}\right){\left[C_{c}\right]}_{\infty }=Y_{c}K_{Q_{t}}{\left[C_{t}\right]}_{\infty }\;$$

### 2.5. Singlet-Oxygen-Induced CR Isomerization

#### 2.5.1. Methanolic Solutions

In the reaction between crocin and singlet oxygen generated by the thermal dissociation of the 1,4-endoperoxide of 3-(4-methyl-1-naphthyl)proprionic acid (EP) in MeOH, we observed difference absorbance spectra, determined as absorbance at time t of the reaction at 35 °C minus the absorbance at time 0, rather similar to those obtained by irradiation (Figure 5). The spectra were corrected for the EP absorption spectrum (Appendix A) that partially overlaps to the CR absorbance spectra in the UV range. Difference absorbance at the isosbestic point (around 380 nm) remained constant during the reaction, indicating that bleaching reactions were negligible during the measuring time adopted.

As in the case of photoisomerization, the *all-trans ↔ cis* conversion could be followed by the change in the DA at the maximal molar extinction difference at 463 nm. Using the second peak of the DA at 321 nm is more difficult because of the partial overlapping of the EP absorption. Simultaneous measurement of the absorbance at 266 nm (see evolution in Appendix A) was used to calculate the release of ^1^O_2_ that appeared almost linear with time, even if the best fit of data required an exponential function.

Figure 6 shows the time evolution of the DA spectra peak values at 463 nm of methanolic solutions during the reaction with ^1^O_2_ of both all-trans- and 13-cis-CR.

Because of the absence of significant bleaching, Equations (22) and (23) modify as follows:(25)$$Y_{c}=\dot{-C_{t0}}\cdot \frac{K_{s}+K_{Q_{t}}C_{t0}}{K_{Q_{t}}C_{t0}\cdot K_{D}EP_{0}Y_{SO}}$$
(26)$$Y_{t}=\dot{-C_{c0}}\cdot \frac{K_{s}+K_{Q_{c}}C_{c0}}{K_{Q_{c}}C_{c0}\cdot K_{D}EP_{0}Y_{SO}}$$

The rate constants of the ^1^O_2_ physical quenching in MeOH were previously determined by the Young’s method [38] and were $K_{Q_{t}}=$7.37 × 10^9^ M^−1^ s^−1^ and $K_{Q_{c\;}}=$6.46 × 10^9^ M^−1^ s^−1^ for all-trans- and 13-cis-CR, respectively (Speranza, personal communication).

By measuring the initial CR conversion rate from the exponential curve fitting of the kinetics of the DA_463_ (Figure 6) and the initial EP concentration, we could calculate the relative yields of isomerization from the triplet state.

These data for both the *all-trans → cis* ($Y_{c}$) and the *cis → all-trans* ($Y_{t}$) reactions are reported in Table 2, as measured (m) and predicted (*p*) values.

The $Y_{c,m}$ was about half the yield measured for all-trans isomerization starting from 13-cis-CR.

Following the reaction until the steady state, the equilibrium relative concentration of 13-cis-CR starting from all-trans was calculated to be about 20% (Table 2). At this stage, $\dot{C_{t}}=0$, and the Equation (24), with $K_{R_{t}}$ and $K_{R_{c}}$ negligible, give
(27)$$Y_{c}K_{Q_{t}}C_{t\infty }=Y_{t}K_{Q_{c}}C_{c\infty }$$
that makes it possible to predict the yields $Y_{t}$ for the back isomerization to be almost five times higher than the measured $Y_{c,m}$ (Table 2).

The 13-cis-CR concentration of equilibrium was more than double starting from 13-cis-CR with respect to that obtained by starting from all-trans-CR, while the predicted $Y_{c,p}$ was not significantly different than the measured $Y_{c,m}$.

#### 2.5.2. Aqueous Solutions

The analysis of the ^1^O_2_-induced isomerization of crocin in water solutions is complex because of the presence of non-negligible bleaching reactions. This process was evidenced by the HPLC data of samples processed at different times of the reaction.

In Figure 7A, the decrease with time in the total HPLC area of compounds detected for both the all-trans- and 13-cis-CR reactions with ^1^O_2_ is reported, as an index of the bleaching of compounds. However, at the same time, the process of isomerization still occurred, as recognized by the kinetics of the relative isomer concentrations shown in Figure 7B,C for the all-trans to cis and the cis to all-trans conversions, respectively.

As in the case of the MeOH solutions, to calculate the isomerization yields from Equations (22) and (23), we need to know the chemical and physical ^1^O_2_-quenching rate constants for the two crocin isomers in the aqueous medium. The rate constants, $K_{Q_{t}}$ and $K_{R_{t}}$, for the all-trans-CR were previously reported [23]; however, the reaction model considered in that study was not complete since the all-trans to 13-cis isomerization was neglected.

On the other hand, the chemical bleaching reaction was considered to occur when starting from all-trans-CR but not when starting from 13-cis-CR.

We have the evidence that both bleaching and isomerization processes are present whatever the starting reaction CR isomer is and, therefore, the rate constant values previously reported must be revised.

Our HPLC analysis of the reactions made it possible to determine the rate of compound disappearance by fitting the change with time of the total chromatograph peak areas (Figure 7A), as well as the rate of isomerization by fitting the change with time of the isomer-relative concentrations (Figure 7B,C).

As a result, for the all-trans-CR reaction with ^1^O_2_, the isomerization rate *(*$Y_{c}K_{Q_{t}})$ was 1/3 of the rate of bleaching ($K_{R_{t}})$, leading to the recalculated rate constants as
$$K_{R_{t}}=0.42\;\cdot \;10^{8\;}M^{-1}s^{-1}\;\mathrm{and}\;K_{Q_{t}}=1.81\;\cdot \;10^{9\;}M^{-1}s^{-1}$$

For the 13-cis-crocin reaction with ^1^O_2_, assuming a rate of bleaching ($K_{R_{c}})$ similar to that calculated for all-trans-CR, the resulting physical quenching rate constant was
$$K_{Q_{c}}=1.85\;\cdot \;10^{9\;}M^{-1}s^{-1}$$

According to the revised rate constants, we calculated the relative yields of isomerization from the triplet state for both the all-trans → cis ($Y_{c}$) and the cis → all-trans ($Y_{t}$) reactions, as reported in Table 2.

The $Y_{c}$ was 5.5 times lower than the yield measured for all-trans isomerization starting from 13-cis-CR and about one order of magnitude lower than the $Y_{t}$ predicted by the model (Equation (24)).

The measured $Y_{c}$ was also significantly (*p* = 0.002) lower than the predicted $Y_{c}$ starting from 13-cis-CR.

The 13-cis-CR concentration of equilibrium was significantly (*p* = 0.048) higher starting from 13-cis-crocin with respect to that obtained by starting from all-trans-crocin.

Following the ^1^O_2_-CR reaction by the HPLC analysis, it is evident that a third compound, called x-cis-CR, appeared and increased with time (Figure 7). Its amount was limited to about 1.5% of the total compounds when the reaction was started from all-trans (Figure 7B), while it reached about 7% in the 13-cis-CR reaction with ^1^O_2_ (Figure 7C).

## 3. Discussion

In this work, we performed a complete analysis of the photon and ^1^O_2_ induction of the cis–trans isomerization in the water-soluble carotenoid CR. The proposed spectrophotometric method allowed the simultaneous monitoring of the configurational change, possible bleaching and the ^1^O_2_ production during the whole process.

Photoisomerization was followed at two absorption bands, that is, one close to the second main peak of absorption (463 nm) and the other at the “cis” absorption peak (321 nm). During the photoisomerization, absorbance at these two bands changed in an opposite direction and depending on the starting molecule (Figure 3, Figure 4 and Appendix A). The process was reversable and concerned essentially only two isomers, the all-trans- and 13-cis-CR, as proved by the presence of an isosbestic point in the DA spectra, without any significant bleaching. If other cis isomers were involved, they should have been produced at rather low amounts and should have had absorption properties similar to the 13-cis-CR. 

The presence of the single 13-cis-isomer means that this is the most stable configuration with the highest rotational energy barrier to switch to all-trans. The other possible cis isomers are not formed at significant amounts or are rapidly isomerized back to all-trans because of lower rotational barriers, to be confirmed with theoretical studies similar to those applied to β-carotene and lycopene cis–trans isomerization [39].

Interestingly, results in term of isomerization quantum yields and isomer photoequilibrium concentrations obtained in both water and methanolic solutions were rather similar, and closely recall that described for β-carotene in non-polar solvents [16,18,19].

It is likely that the cis–trans isomerization occurs as relaxation of the CR triplet excited state through a twisted intermediate state common to the all-trans and 13-cis isomers. 

The much larger (>10-times) $\phi_{t}$ with respect to $\phi_{c}$ suggests the presence of a higher rotational energy barrier from the all-trans triplet state than from the 13-cis-CR, analogously to the triplet potential energies calculated for β-carotene and lycopene [39].

The involvement of the carotenoid triplet state in the isomerization process is confirmed by the photoisomerization results obtained in the methanolic solution saturated with molecular oxygen (Table 1). In fact, oxygen is known to induce inter-system crossing [40] that enhances $\phi_{c}$ and $\phi_{t}$ by about 2- and 5-fold, respectively, with respect to those in argon-saturated solutions, without appreciable bleaching.

No data about the quantum yield for photoisomerization of crocinoids are reported in the literature. However, Craw and Lambert [33] indicated a quantum yield of triplet formation of crocetin of less than 10^−3^ in oxygen-free aqueous solutions. This is due to the competition between the internal conversion and intersystem crossing in the singlet excited-state relaxation process, in favor of the internal conversion. Accordingly, our value of $\phi_{c}$ that is about an order of magnitude lower (Table 1), is consistent with a photoisomerization pathway involving the excited triplet state. 

Analyzing the CR-^1^O_2_ interaction in methanolic solutions, we still observed the all-trans↔cis reversable conversion with spectral absorbance changes similar to those generated by photoisomerization (Figure 5 and Figure 6). In water solutions, however, a certain amount of compound bleaching appeared along with the isomerization process.

It is generally accepted that physical quenching is the primary way carotenoids reduce singlet oxygen [11]. As depicted in Figure 2, once a CR isomer reaches its triplet state, produced by the energy transfer from ^1^O_2_, it can decay back to the starting configuration or rotate around the C13=C14 double bond, leading to the twisted configuration. The measured yield of cis→all-trans isomerization from the triplet state, $Y_{t,m}$, was about 1.9 (MeOH) and 5.5 (H_2_O) times higher than $Y_{c,m}$, confirming again the easier pathway for the cis to trans rotation with respect to the opposite, analogously to what happens in the photoisomerization.

In MeOH solutions, we did not observe any significant bleaching occurring during the CR-^1^O_2_ reaction. This is in accordance with the literature, showing that the ^1^O_2_ chemical quenching by carotenoids occurs to a much lower extent with respect to physical quenching [3,22,24]. Even in a reverse micelle system, used to study the medium heterogeneity effect on the ^1^O_2_ quenching rate constants, K_Q_ for different carotenoids was about four orders of magnitude higher than K_R_ [27].

In the characterization of the CR isomerization, it is also essential to compare the ^1^O_2_-quenching ability of the cis and trans isomers. The rate constants of the ^1^O_2_ physical quenching in organic solvents for both CR isomers were found to be lower than those of all-trans- and cis-β-carotene, in accordance with their lower number of conjugated double bonds [15]. The difference in K_Q_ between all-trans- and 13-cis-CR was around 15%, as was observed between all-trans and 15-cis-β-carotene [15]. Accordingly, in aqueous solutions, we estimated quite similar values of $K_{Q_{t}}$ and $K_{Q_{c}}$.

Some authors [41,42] have suggested that the cis geometrical configurations of astaxanthin possess higher antioxidant properties with respect to the all-trans configuration. Although a clear mechanism for that has not been found yet, it is possible that this difference is related to the more efficient isomerization from cis to all-trans than vice versa, as we observed in the CR-^1^O_2_ reaction, that can promptly quench ROS without any significant chemical consumption of molecules.

For CR in H_2_O, we found that for both isomers, K_Q_ was at least 40 times higher than K_R_. However, in this medium, the disappearance of the starting compound was not negligible (Figure 7A).

The photon- and ^1^O_2_ -induced-reactions model we used properly described the isomerization processes from all-trans- to 13-cis-CR. However, when starting the reactions from the 13-cis isomer, some discrepancies in terms of absolute values of quantum yields, triplet state decay yields and relative concentrations of isomers with respect to the process starting from all-trans-CR were found. 

We do not have a conclusive explanation for that. The observed inconsistency can be due to the limited purity of the starting 13-cis solutions and/or to the involvement in the isomerization processes of more than the two all-trans and 13-cis compounds. Indeed, in the H_2_O medium, clear evidence of the ^1^O_2_-induced production of an additional molecule was reported (Figure 7C). It could not be definitely identified; however, it is likely to be another cis or a di-cis-isomer of CR.

This explanation should be in accordance with the isomerization patterns and quantum yields proposed by Kuki et al. [16] for the triplet-sensitized and direct photoisomerization of β-carotene. In synthesis, the trans to cis isomerization is mainly favored towards the central cis-isomers; the cis to all-trans isomerization is the far more efficient process for the central cis-isomers but with significant chances to produce other cis- or di-cis-isomers.

## 4. Materials and Methods

### 4.1. Chemicals and Solutions

Crocin was extracted and purified from commercially available saffron according to Speranza et al. [35]. The isolation of the all-trans and 13-cis configurational isomers was achieved by preparative HPLC [35]. Di-n-octylamine for HPLC was from Aldrich-Chemie, Germany, and other chemicals were analytical- or HPLC-grade reagents. Phosphate buffer solutions were purged with Ar. 

The spectral extinction coefficients of all-trans-CR (ε_t_) in both methanolic and aqueous solutions were measured directly. For 13-cis-crocin, since its HPLC analysis showed the content of about 20% of unidentified impurities, the ε_c_ spectrum was indirectly estimated from the difference absorbance spectrum of an all-trans-CR solution irradiated to the photoequilibrium and measuring the $C_{c}\left(\infty \right)$ and $C_{c}\left(\infty \right)$ isomer concentrations by HPLC.

Solutions of *all-trans-* and 13*-cis-*CR were prepared by dissolving small amounts of the pigment (0.3–0.5 mg) directly in methanol or 0.1 M sodium phosphate buffer (pH 7.4). They were either deoxygenated by Ar bubbling or aerated (saturated by O_2_ bubbling).

Fresh crocin solutions were prepared each day and manipulated under dim red light.

### 4.2. Monitoring of Photoreactions

Samples of CR solutions (volume 0.44 mL), with concentrations ranging from 5 µ*M* to 15 µ*M*, were irradiated (argon laser, in the cell compartment of a spectrophotometer (PYE UNICAM model 8800) at the 488 nm Ar laser (Spectra-Physics model 165-008) line. An unirradiated sample was maintained in the reference position in order to record differential absorption values. Variations of the differential absorbance during irradiation were simultaneously monitored at 321, 463 and 380 nm, corresponding respectively to the negative and positive maximum and to the isosbestic point of the differential absorption spectrum of the all-*trans*/*13-cis* CR mixture. The 380 nm wavelength was an isosbestic point, so that its variation could be considered an index of the presence of photobleaching or the photoproduction of other compounds in addition to all-trans- and 13-cis-CR. A 2 × 10 mm quartz cuvette was used, with the thinner side exposed to the laser light to minimize light attenuation across the solution. The laser beam was expanded to provide essentially uniform energy irradiance across the cuvette face. 

The photon fluence rate at the cuvette surface was about *F* = 1.9 × 10^16^ photons cm^−1^ s^−1^ (laser power meter: EG&G model 460, instrumental accuracy = 5%). Measurements were carried out at 26 ± 2 °C.

### 4.3. Singlet Oxygen Production and Reaction Monitoring

Singlet molecular oxygen was produced by the thermal dissociation of 3-(1,4-epidioxy-4-methyl-1,4-dihydro-1-naphthyl)propionic acid (EP) to produce naphthalenic acid (Naph). 

The ^1^O_2_ concentration with time could be determined by measuring the evolution of the Naph concentration corrected for the yield of ^1^O_2_ formation, $Y_{SO}$, that was earlier estimated to be more than 82% in methanol [43] and 45% in H_2_O [44].

The Naph concentration was measured spectrophotometrically by the absorbance at 266 nm (isosbestic point for the all-trans-↔cis-CR isomerization) according to an exponential function defined as
(28)$$A_{Naph}^{266}\left(t\right)=\varepsilon_{Naph}^{266}{\left[EP\right]}_{0}\left(1-e^{-K_{D}t}\right)$$
where $\varepsilon_{Naph}^{266}$ is the molar extinction coefficient of Naph at 266 nm, ${\left[EP\right]}_{0}$ is the initial concentration of the endoperoxide, $K_{D}$ is the decomposition rate constant of the endoperoxide. This method was previously used to calculate the ^1^O_2_-quenching rate constants of anthracyclines, bixinoids and crocinoids [23,25,37]. It is advantageous with respect to photochemical ^1^O_2_ production since it avoids the direct cis–trans photoisomerization of carotenoids.

Two separate solutions of crocin (about 15 µM) and EP (about 5 mM) in methanol or phosphate buffer were mixed at 0 °C. At this temperature the decomposition of EP to Naph is negligible [43]. The resulting solution was deoxygenated by Ar bubbling and then rapidly heated to 35 °C by a Peltier device and maintained at this temperature (±0.1 °C) inside the spectrophotometer. 

The time evolution of the CR-^1^O_2_ reaction was monitored following the temporal changes of the absorbance at 463 nm, the maximum of the isomerization difference absorption spectrum. The bleaching of crocin was simultaneously checked by measuring absorbance at 380 nm, isosbestic point between the two crocin isomers. The overlapping of the Naph absorbance with that of the cis/trans-CR mixture in the ultraviolet region below 340 nm made it impossible to follow the evolution of the “cis peak” at 321 nm. 

### 4.4. HPLC Analysis

Crocin solutions were analyzed by an isocratic reversed-phase HPLC using a Waters Liquid Chromatograph fitted with a model 740 integrator, a 5 µm, 25 × 0.46 cm Ultrasphere ODS column (Beckman, CA, USA) or Ultremex ODS column (Phenomenex, Torrance, CA, USA) and a 4.6 × 0.45 cm precolumn. The eluent was a mixture of water and 0.1 M di-n-octylamine acetate in MeOH (1:2 *vol/vol*) at a flow rate of 1 ml min^−1^.

Before injection, samples (40 mL) from the aqueous solutions were diluted with 0.1 M di-n-octylamine acetate in MeOH (80 mL); samples from the MeOH solutions were mixed with one equal volume of H_2_O and one of 0.1 M di-n-octylamine acetate in MeOH. This precaution avoided undesirable peaks in the chromatograms due to solvent interferences. The mixtures were centrifuged and 50 mL were injected into the column. 

Peaks were monitored at 440 nm (absorption maximum) and identified by reference to standards. Cis isomer peak areas were corrected for the difference in the extinction coefficient at 440 nm in the HPLC mobile phase with respect to all-trans (a factor of 1.36).

### 4.5. Data Analysis

Statistical analysis and curve fitting were carried out with the SigmaPlot for Windows Version 14.0 software (Systat Software, Inc., San Jose, CA, USA). Comparison of mean values was performed according to the Welch’s *t*-test, with *p* values < 0.05 considered statistically significant. The coefficient of determination, R^2^, was used to estimate the curve fitting quality.

## 5. Conclusions

In summary, the present study provides a new insight on the isomerization processes occurring in the water-soluble carotenoid crocin induced by light or by the physical quenching of ^1^O_2_. Although the presence of a significant chemical quenching of ^1^O_2_ in plants was suggested [45], the cis–trans isomerization process can represent an important physiological role in vivo as a tool to quench reactive oxygen species with a limited requirement of regeneration of the carotenoid. Further investigations by time-resolved spectroscopy are needed to better and conclusively characterize the whole isomerization process in the water-soluble crocinoids.

## Figures and Tables

**Figure 1 ijms-24-10783-f001:**
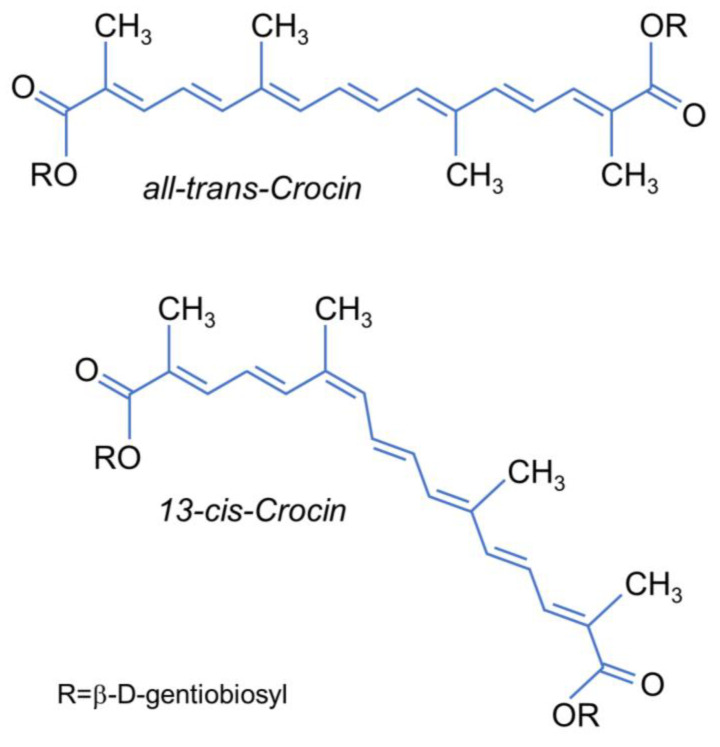
Chemical structure of all-trans- and 13-cis-crocin.

**Figure 2 ijms-24-10783-f002:**
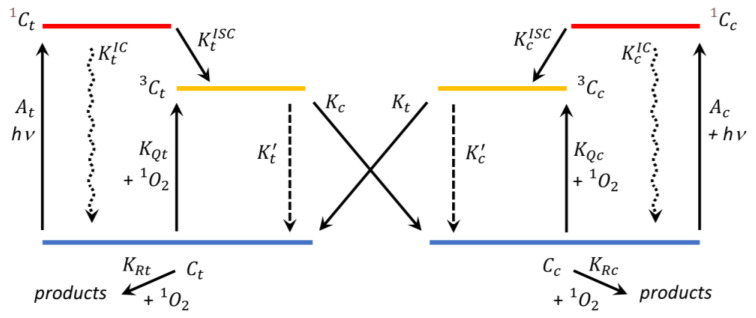
Scheme of the main photochemical and ^1^O_2_-induced trans–cis isomerization processes in crocin. $A_{t}$ and $A_{c}$ are the excitation rate constants from the steady state of all-trans-CR ($C_{t}$) and 13-cis-CR ($C_{c}$), respectively, to their singlet excited states, C1t and C1c, respectively. The $K_{t}^{IC}$ and $K_{c}^{IC}$ are the rate constants for the internal conversion from singlet excited- to ground-states; the $K_{t}^{ISC}$ and $K_{c}^{ISC}$ are the rate constants for the intersystem crossing to generate the triplet energy states, C3t and C3c of $C_{t}$ and $C_{c}$, respectively. $K_{Q_{t,c}}$ and $K_{R_{t,c}}$ are the bimolecular rate constants for physical and chemical quenching of ^1^O_2_, respectively. The deexcitation rates of the CR triplet states are indicated by $K_{t}$ and $K_{c}$ for the trans↔cis isomerization processes and by $K_{t}^{’}$ and $K_{c}^{’}$ for the internal relaxation processes.

**Figure 3 ijms-24-10783-f003:**
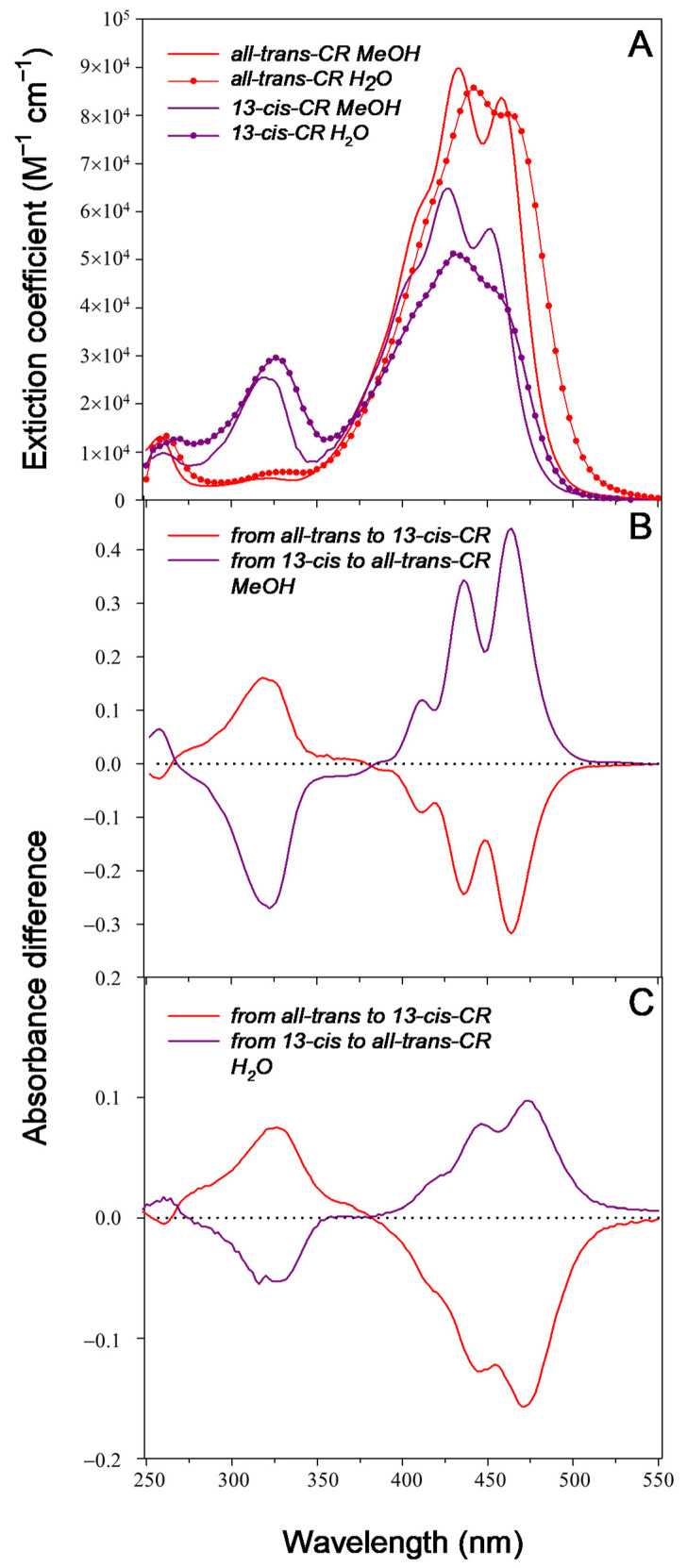
Extinction coefficients of all-trans-CR and 13-cis-CR in methanol and water (**A**). Difference (irradiated minus unirradiated) absorbance spectra recorded after irradiation at 488 nm to the photoequilibrium of both solutions of all-trans- (dashed red line) and 13-cis-CR (purple line) in methanol (**B**) and water (**C**) on a 1 cm pathlength. The initial concentration was 15 µM and 5 µM for all-trans- and 13-cis-CR solutions, respectively.

**Figure 4 ijms-24-10783-f004:**
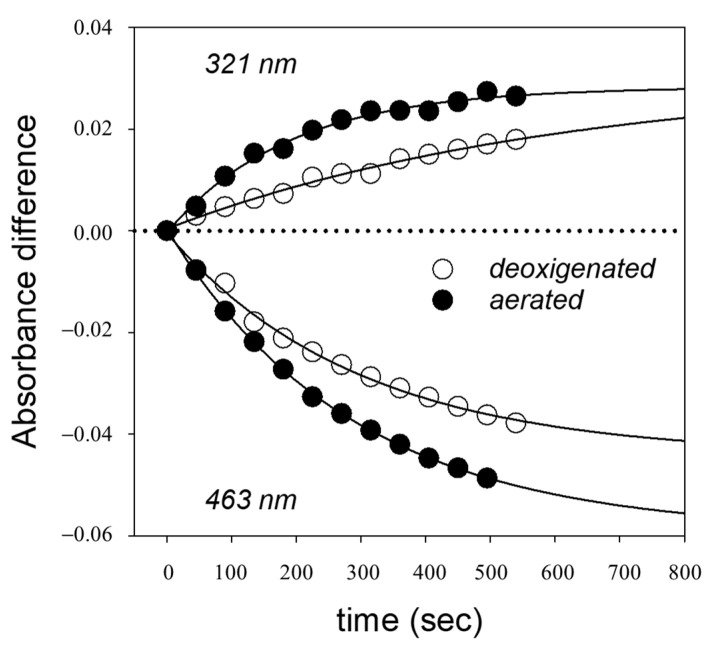
Time evolution of the difference absorbance values at 321 and 463 nm during irradiation at 488 nm, starting from a solution of all-trans-CR in methanol at the concentration of 7.25 µM. Full line represents the data-fitting exponential curves.

**Figure 5 ijms-24-10783-f005:**
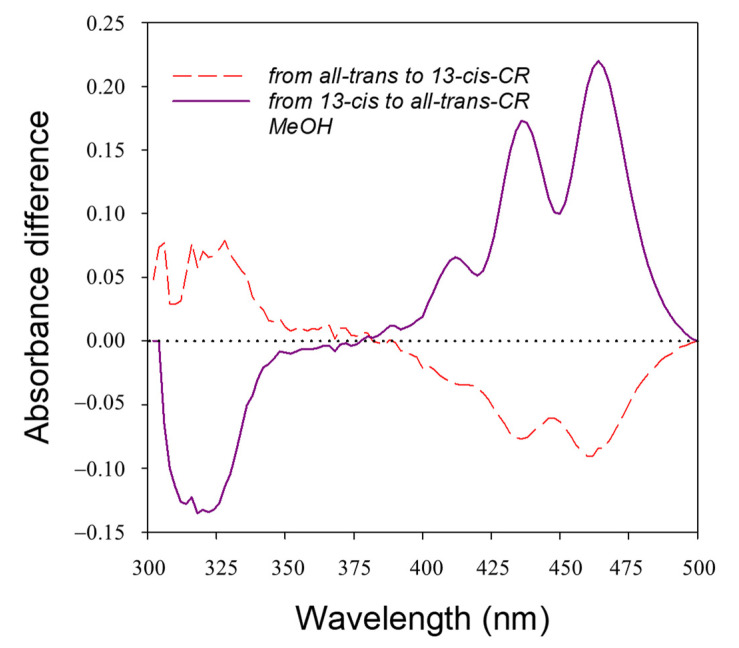
Difference absorbance spectra between a methanolic solution of CR and EP after 30′ of reaction at 35 °C with respect to the time 0 solution, starting from all-trans- (dashed red line) and 13-cis-CR (purple line) solutions, both with an initial concentration of 10.5 µM.

**Figure 6 ijms-24-10783-f006:**
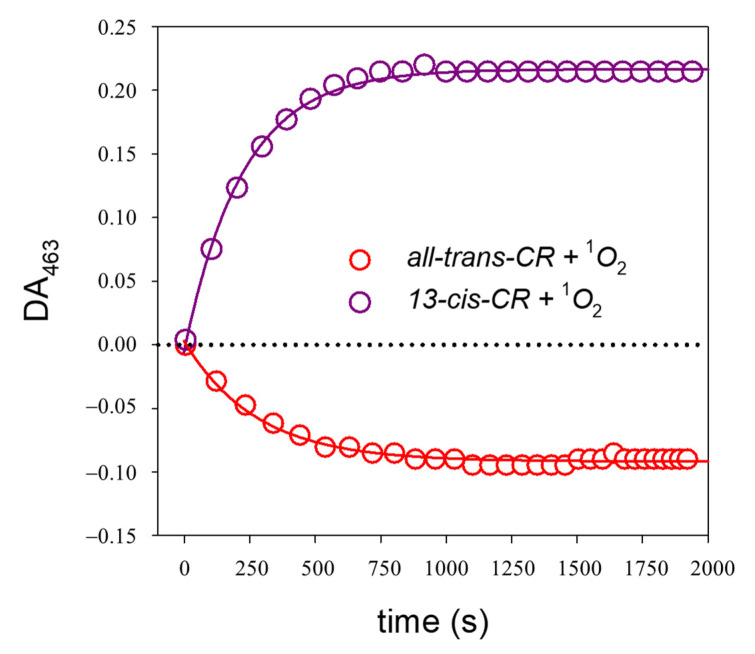
Time evolution of the difference absorbance values at 463 nm during the reaction of CR with ^1^O_2_, starting from a solution of all-trans-CR (red line) or 13-cis-CR (purple line) in methanol, both at the concentration of 10.5 µM. Full line represents the data-fitting exponential curves.

**Figure 7 ijms-24-10783-f007:**
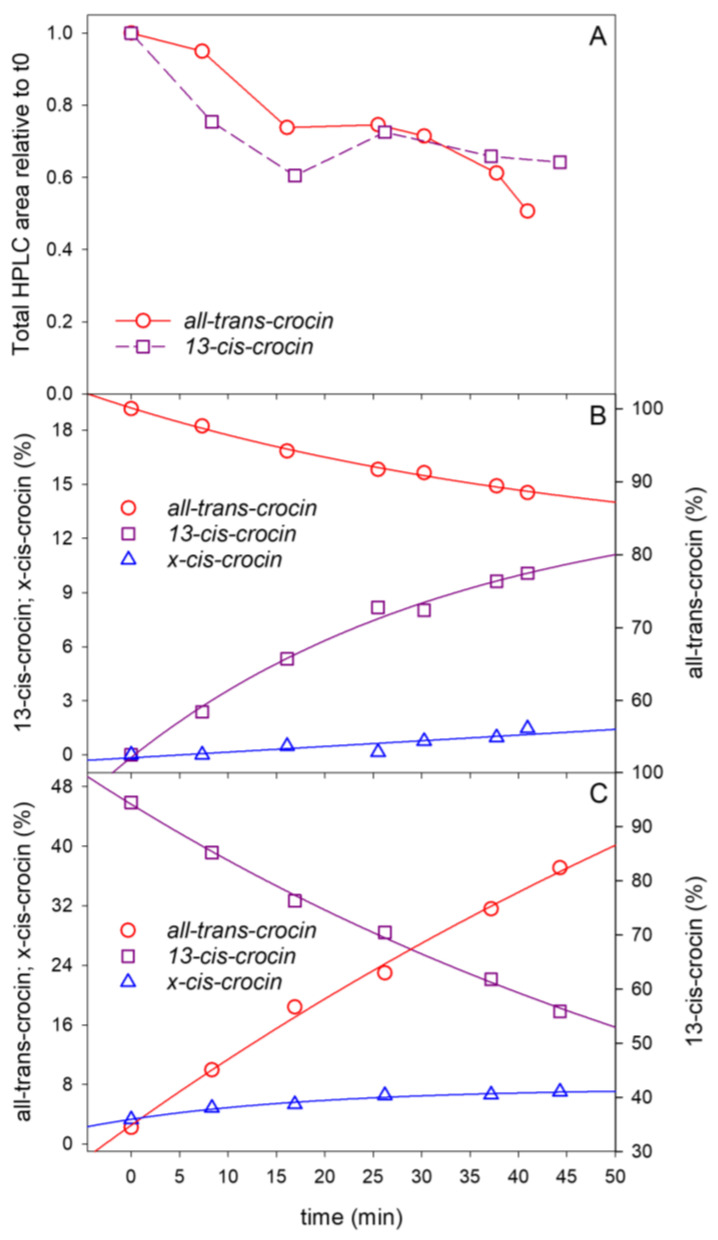
Variation of the total HPLC area of crocin compounds with time of their reaction with ^1^O_2_ (**A**). Kinetics of the relative concentration of crocin isomers induced by the reaction with ^1^O_2_ starting from all-trans-crocin (**B**) or from 13-cis-crocin (**C**). The solid lines in B and C represent the exponential function fitting of data.

**Table 1 ijms-24-10783-t001:** Average values (±SD) of quantum yields of trans ↔ cis photoisomerization and 13-cis-CR photoequilibrium concentration under laser irradiation at 488 nm.

	$\phi_{c,m}$	Trans → Cis $\phi_{c,p}\;$ (Predicted by Equation (12))	$C_{c}\left(\infty \right)$ (%)	$\phi_{t,m}$	Cis → Trans $\phi_{t,p\;}$ (Predicted by Equation (12))	$C_{c}\left(\infty \right)$ (%)
MeOH Argon saturated	4.75 ± 2.10 × 10^−4^	3.14 ± 1.23 × 10^−4^	17.6 ± 1.7	2.10 ± 0.76 × 10^−3^	6.31 ± 2.71 × 10^−3^	30.8 ± 9.7
MeOH Oxygen saturated	1.11 ± 0.37 × 10^−3^	1.65 ± 0.62 × 10^−3^	19.3 ± 2.5	1.16 ± 0.29 × 10^−2^	1.30 ± 0.26 × 10^−2^	29.4 ± 11.9
H_2_O Argon saturated	2.6 ± 0.6 × 10^−4^	1.8 ± 0.7 × 10^−4^	17.0 ± 2.8	2.4 ± 0.4 × 10^−3^	6.2 ± 1.4 × 10^−3^	26.2 ± 4.1

**Table 2 ijms-24-10783-t002:** Average values (±SD) of yields of *all-trans ↔ cis* isomerization and 13-cis-CR equilibrium concentration induced by the reaction with singlet oxygen.

	$Y_{c,m}$	*all-trans**→ cis* $Y_{c,p}$ Predicted	$C_{c}\left(\infty \right)$ (%)	$Y_{t,m}$	*cis**→ all-trans* $Y_{t,p}$ Predicted	$C_{c}\left(\infty \right)$ (%)
MeOH	0.113 ± 0.009	0.180 ± 0.038 *	20.5 ± 0.3	0.215 ± 0.031	0.501 ± 0.048 *	48.7 ± 2.0
H_2_O	0.058 ± 0.009	0.115 ± 0.017 **	13.9 ± 1.3	0.318 ± 0.048	0.1650 ± 0.073 **	23.0 ± 4.0

* From Equation (27); ** From Equation (24).

## Data Availability

Not applicable.

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
