# Peer review of "Photon- and Singlet-Oxygen-Induced Cis–Trans Isomerization of the Water-Soluble Carotenoid Crocin"

_ijms, 2023, doi:10.3390/ijms241310783_

Round 1

Reviewer 1 Report

The research paper entitled "Investigation of Photon and Singlet Oxygen Mediated Cis-Trans Isomerization of the Water-Soluble Carotenoid Crocin" written by Fusi and coworkers, focuses on the HPLC/UV evaluation of the isomerization process of the carotenoid crocin, induced by singlet oxygen quenching or visible light irradiation exposure. The study provides new insights into the understanding of the tested compound isomerization process.

The introduction presents relevant aspects of the subject matter, and the study is appropriately designed. Detailed results and comprehensive descriptions of the experimental protocols are included. While the scientific significance of the presented data may not be astounding, the manuscript aligns with the scope of the International Journal of Molecular Sciences (IJMS). I recommend making minor textual corrections and support the publication of this work.

Author Response

The authors would like to thank both reviewers for their constructive comments and suggestions. The text has been modified accordingly and the modifications have been highlighted along the manuscript.

On behalf of all the other authors,

Kind regards,

Giovanni Romano

-----

Answer to the reviewer

We have revised the whole manuscript and made textual corrections, as detailed along the text, both to improve the overall manuscript quality and clarity and to correct minor issues due to incorrect data presentation. A brief Conclusions section was introduced.

Reviewer 2 Report

The manuscript of Fusi et al. describes the investigation of the cis-trans isomerization reactions of crocin induced by light or reactions with singlet oxygen. The authors use UV/vis spectophotometry, high-performance liquid chromatography, and a kinetic analysis. The topic is interesting, the study is detailed, and the manuscript is well-written (with exception of several points indicated below). Therefore, I recommend this manuscript for publication (after addressing the comments below).

1) Could the authors provide more reasoning why they exclude isomerization following internal conversion from the singlet excited state to the singlet ground state, and why they assume that the intersystem crossing outcompetes the internal conversion in the case of crocin?

2) Fig. 4 is for the trans-->cis isomerization. Since the authors also report results on the cis-->trans isomerization, it would be good to provide a similar figure for the cis-->trans reaction (either in the main text or in the supporting information).

3) In Tables 1 and 2, I think that for the trans-->cis isomerization, the correct subscript is "c", and for the cis--> trans it is "t", i.e.

phi_c,m; phi_c,p; C_c(inf) for trans-->cis

phi_t,m; phi_t,p; C_t(inf) for cis-->trans

(in Tab. 2, the same comment holds for Y instead of phi)

Am I correct? If not, please elaborate more on this point.

4) In Eq. (9), the two equations are identical; please correct.

5) Page 16, line 424: "The measured yield of all-trans→cis isomerization from the triplet state, Y t , was about 5
(MeOH) and 10 (H2O) times higher than Y c , "

Should not it be cis-->trans? (otherwise, this sentence contradicts the abstract, Line 21)

Should not it be phi instead of Y? Please refer here to the corresponding table (I think it is Tab. 1).

6) Page 4: "Equations (1) are valid in the limit of optically thin solution, αl ≪ 1,"

And also Eq. (2)? Could the authors comment why they introduce the spatial averaging for F?

7) On Page 9, "P" appears in "no significant difference between the average values calculated at
the two spectral bands (P > 0.5) was observed"

It would be good to explain what this P stands for.

8) There are many one-sentence paragraphs in the main text. For example on Page 15:

"The photoisomerization of CR in both methanolic and aqueous media closely recall that described for -carotene in non-polar solvents [16,18,39].

Interestingly, results in term of isomerization quantum yields and isomer photoequilibrium concentrations obtained in water solutions were rather similar to those obtained in methanolic solutions.

It is likely that the cis-trans isomerization occurs as relaxation of the CR triplet excitedstate through a twisted intermediate state common to the all-trans and 13-cis isomers."

and also in Introduction.

I think that they should be combined in bigger paragraphs (where applicable).

9) Page 8: " Furthermore, the pho-
toisomerization reaction was faster in the presence of oxygen than without oxygen."

It would be good to provide some explanation here.

10) Page 2: "Since 1O2 possesses a relatively low energy level (94.5 kJmol -1)"

94.5 kJmol -1 with respect to what? Please explain.

11) It would be good to add a short "Conclusions" section.

Abstract, line 21: that --> than

Abstract, line 15: please define abbreviation "HPLC"

Abstract, line 23: please define abbreviation "ROS"

Line 31: undergo to --> undergo

Line 45: are also be involved --> are also involved

Line 57: and reacts --> and it reacts (?)

Line 118: 1O2 --> 1O2

Line 129: lambdai --> lambdai

Lines 137 and 141: F --> F

Line 365: call-cis --> cis (?)

Line 498: imultaneously --> simultaneously
